# FastDCFlow: Fast and Diverse Counterfactual Explanations Using Normalizing Flows

## Abstract

Machine-learning models, which are known to accurately predict patterns from large datasets, are crucial in decision-making. Consequently, counterfactual explanations—methods explaining predictions by introducing input perturbations—have become prominent. These perturbations often suggest ways to alter predictions, leading to actionable recommendations. However, the current techniques require resolving the optimization problems for each input change, rendering them computationally expensive. In addition, traditional encoding methods inadequately address the perturbations of categorical variables in tabular data. Thus, this study propose "FastDCFlow," an efficient counterfactual explanation method using normalizing flows. The proposed method captures complex data distributions, learns meaningful latent spaces that retain proximity, and improves the predictions. For categorical variables, we employed "TargetEncoding," which respects ordinal relationships and includes perturbation costs. The proposed method outperformed existing methods in multiple metrics, striking a balance between trade offs for counterfactual explanations.

## 1 Introduction

Machine learning (ML) technologies strive to replicate the learning capabilities of computers. With recent advancements, ML can now address decision-making tasks that were traditionally determined by humans. Such decisions often require counterfactual explanations (CE). Counterfactuals envision unobserved hypothetical scenarios. For instance, if a bank's algorithm denies a loan, a counterfactual might reveal that an extra \$3,000 in annual income would have secured approval, guiding the applicant towards future success.

When implementing CE in ML, methods often introduce input perturbations to optimize the target variable predictions. These altered inputs produce counterfactual samples (CF), representing unobserved scenarios. As outlined by Wachter et al. (2017), CFs should satisfy two constraints that are often in a trade-off relationship: validity and proximity. Validity ensures that perturbations modify inputs such that ML models yield the desired output, such as changing loan denial to approval in two-class classifications. Whereas, proximity requires the perturbations to remain as close as possible to the original input. However, relying solely on these two metrics can be misleading, as models may appear to perform well while generating nearly identical CFs. Given that they are not necessarily unique, assessing the diversity among CFs is also crucial. Interestingly, as reported by Pawelczyk et al. (2022), CF generation echoes the principles of adversarial attacks (Szegedy et al., 2013; Ballet et al., 2019), wherein inputs are perturbed until the predicted class shifts, typically when the binary prediction probability reaches $\geq 0.5$. However, in contrast to adversarial attacks, which focus on threshold shifts, we advocated CFs that span various prediction probabilities.

The primary goal of the CE is to generate a broad range of diverse CFs that adhere to both validity and proximity. However, there are two challenges with the current methods. The first is the presence of both categorical and continuous variables in the tabular data. For instance, in the context of a loan application, suggesting a counterfactual transition from a "high school graduate" to a "PhD holder" may be impractical. If a counterfactual suggests upgrading to "college graduate," this could be a more realistic change for approval. Current preprocessing methods, such as OneHotEncoding (OHE) and LabelEncoding (LE), as in Mothilal et al. (2020); Duong et al. (2023), tend to produce unrealistic outcomes. OHE often misinterprets categorical shifts, equating a jump from high school

to PhD with a change to college. With LE, perturbations become non differentiable, which is unsuitable for deep-learning models. The second challenge is related to the overhead of crafting individual CFs for each input. As described in Rodriguez et al. (2021), generating CFs often involves the optimization of variables tailored to each input, which results in inefficiencies as the volume of inputs expands and impacts storage and execution time.

To address these challenges, we introduced fast and diverse CEs using normalizing flows (FastD-CFlow). Within FastDCFlow, we managed the perturbations of categorical variables using TargetEncoding (TE). This method aggregated the average value of the target variable for each categorical level using training data and substituted the categorical values with their corresponding averages. Thus, categorical variables were effectively transitioned into continuous variables, maintaining a meaningful order of magnitude relative to the predicted values. Furthermore, to generate optimal CFs, we leveraged normalizing flows (Dinh et al., 2014; 2016; Kingma & Dhariwal, 2018) to capture complex data distributions through a reversible latent space. In summary, Our main contributions include: (1) Using TE, we lower the model's learning cost and diversify the generated CFs' predicted values. (2) We introduce FastDCFlow, a method that learns a latent space for ideal CF generation using normalizing flows. We evaluate the CFs against multiple metrics and compare with other methods to confirm our approach's effectiveness.

## 2 RELATED WORK

In this study, we broadly classified the generation algorithms of existing research into two categories: input- and model-based methods.

**Input-based methods.** These methods determine the optimal set of CFs for each unique input, requiring relearning of variables with every input alteration. Wachter et al. (2017) optimized variables for a single CF, employing a gradient descent-based algorithm. Mothilal et al. (2020) introduced DiCE, which was designed to produce multiple CFs. DiCE optimizes sets of multiple CFs by adding an explicit diversity term based on a determinant from the distance matrix between variables. However, the computational complexity increases with variable dimensionality and the number of CFs produced. A genetic algorithm (GA) was used for efficient generation with increasing CF count. Schleich et al. (2021) proposed GeCo, wherein CFs as were conceptualized as genes and crossover and mutation were executed across generations. Dandl et al. (2020) proposed MOC. They framed objective functions through multi-objective optimization capturing trade-offs. Critically, these methods independently optimize the observed variable dimensions, overlooking the impact of unobserved common factors in the data generation process.

When considering the influence of unobserved common factors, one approach perturbs the latent variables presumed to underpin the observed ones. Rodriguez et al. (2021) introduced DIVE, which employed the variational auto-encoder (VAE) (Kingma & Welling, 2013; Rezende et al., 2014; Mnih & Gregor, 2014; Gregor et al., 2014) to determine the optimal perturbations for latent variables. Although DIVE delivers insightful CEs in the image domain, its efficacy on tabular data with categorical variables remains unverified. Many studies have transformed categorical variables into continuous variables using OHE or LE. However, these transformations lack a magnitude relationship between the values. Considering the dimension expansion and sparsity issues, these methods are unsuitable for CE, wherein variable perturbations are paramount. Duong et al. (2023) presented CeFlow, which incorporated normalizing flows into tabular data to learn added perturbations. Although CeFlow adopts variational dequantization (Hoogeboom et al., 2021) to continuously mirror a categorical distribution, it grapples with the absence of explicit ordering for categorical variable values. Consequently, the continuous variables lack proximity to their inherent values.

**Model-based methods.** These approach entail direct learning of the CF generation model using training data. They requires only one learning session. After stabilizing the model with trained parameters, CFs can be readily produced for unobserved inputs (test inputs). Looveren & Klaise (2021) employed a spatial partitioning data structure, the k-d tree (Bentley, 1975), to create a set of training data points as CFs proximate to the test input within the targeted prediction class. However, this search was confined to the scope of observed data points, which rendered the generating of CFs for data not previously observed challenging. Mahajan et al. (2019) introduced a technique that directly learned the latent space for CF generation using VAE. In contrast to VAE as an input-based method that optimized perturbations, this method offered the benefit of eliminating the need

Figure 1: Overview of CFs training and generation using FastDCFlow.

for model retraining for each input. However, because VAE assumes a latent distribution based on continuous variables, certain issues remain in processing data that include categorical variables.

Input-based methods require optimization problems to be solved for each input. This process becomes increasingly inefficient as the number of verification inputs increases. Hence, the proposed FastDCFlow aimed to produce CFs swiftly, adhering to the necessary constraints for tabular data. To achieve this, a model-based approach was employed.

## 3 PROPOSED METHOD

Consider an input space $\mathcal{X}$ with input $\mathbf{x} \in \mathcal{X}$ and its corresponding target $y \in \{0, 1\}$. The ML model $f(\mathbf{x})$ predicts the probability of a positive outcome, where $f(\mathbf{x}) = \hat{y}$ and $\hat{y} \in [0, 1]$. It is assumed that all categorical variables are transformed into continuous variables using TE. The aim of CE is to generate perturbed inputs $\mathbf{x}^{cf}$ that shoud be $f(\mathbf{x}^{cf}) > f(\mathbf{x})$ for the observed input $\mathbf{x}$. To support this, the latent representation $\mathbf{z} \in \mathcal{Z}$ of the input $\mathbf{x}$ was trained. Ideally, $\mathbf{z}$ captures the representation of the unobserved common factors, thereby ensuring both proximity and diversity in the input space.

### 3.1 FASTDCFLOW

We employed normalizing flows to prioritize efficient generation and precise likelihood computation. These flows are generative models that use a function $\boldsymbol{g_\theta}$, parameterized by $\boldsymbol{\theta}$, which transforms the distribution of a dataset into another distribution in the latent space. In the forward transformation, the latent variables $\mathbf{z} \sim p_{\mathcal{Z}}$ are derived by mapping $\boldsymbol{g_\theta} : \mathcal{X} \rightarrow \mathcal{Z}$, transitioning from the input space $\mathcal{X}$ to the latent space $\mathcal{Z}$. The dataset's distribution is transformed back using the inverse $\boldsymbol{g_\theta}^{-1} : \mathcal{Z} \rightarrow \mathcal{X}$, resulting in a random variable $\mathbf{x} \sim p_{\mathcal{X}}$. The probability density function transformation for $\mathbf{x} = \boldsymbol{g_\theta}^{-1}(\mathbf{z})$ is expressed as:

$$\log\left(p_{\mathcal{X}}(\mathbf{x})\right) = \log\left(p_{\mathcal{Z}}(\boldsymbol{g_\theta}(\mathbf{x}))\right) + \log\left(\left|\det\left(\frac{\partial \boldsymbol{g_\theta}(\mathbf{x})}{\partial \mathbf{x}}\right)\right|\right). \tag{1}$$

For effective CF generation, it is essential to produce samples quickly and diversely to enhance the input predictions while preserving proximity. In our proposed FastDCFlow method, beyond precise likelihood computation, we incorporated additional losses, accounting for the constraints that CFs must satisfy. Moreover, to hasten the sample generation, we implemented a model-based method that learned the latent space to enable CF generation in a single training session.

### 3.2 TRAINING AND GENERATION

Figure 1 illustrates the comprehensive framework of the proposed method. Initially, we investigated the training process of FastDCFlow. Subsequently, we explained the approach to foster diversity during test input considerations.

**Minimization of the negative log-likelihood.** The number of training inputs is resresented as $N^{tra}$: Each input pair is denoted by $\mathcal{D} = \{\mathbf{x}_i^{tra}\}_{i=1}^{N^{tra}}$. Owing to the property of normalization flows, which enables the direct evaluation of parameter likelihoods, the negative log-likelihood (NLL) was integrated into the loss function:

$$\mathcal{L}_{nll}(\boldsymbol{\theta}, \mathcal{D}) = -\sum_{i=1}^{N^{tra}} \left(\log\left(p_{\mathcal{Z}}(\boldsymbol{g_\theta}(\mathbf{x}_i^{tra}))\right) + \log\left(\left|\det\left(\frac{\partial \boldsymbol{g_\theta}(\mathbf{x}_i^{tra})}{\partial \mathbf{x}_i^{tra}}\right)\right|\right)\right). \tag{2}$$

**Validity and proximity.** In the process of generating CFs, the initial step involves mapping the input $\mathbf{x}^{tra}$ onto the latent space through $\mathbf{z} = \boldsymbol{g}_\theta(\mathbf{x}^{tra})$. To prioritize sampling in the neighborhood of the input, a perturbed latent variable $\mathbf{z}^*$ is derived by introducing reparametrization trick (Kingma & Welling, 2013) which adds Gaussian noise $\boldsymbol{\epsilon}$ from distribution $\mathcal{N}(\mathbf{0}, \boldsymbol{I})$, resulting in $\mathbf{z}^* = \mathbf{z} + \boldsymbol{\epsilon}$. Then, $\mathbf{x}^{cf}$ is generated from this perturbed variable as $\mathbf{x}^{cf} = \boldsymbol{g}_\theta^{-1}(\mathbf{z}^*)$:

CFs are fundamentally designed to satisfy two primary criteria: validity and proximity to the input. The validity of our model was measured using the cross-entropy loss function, defined as

$$
\begin{aligned}
\mathcal{L}_y(\boldsymbol{\theta}, \mathcal{D}) &= -\sum_{i=1}^{N^{tra}} \left( y_i^{cf} \log \left( f(\mathbf{x}_i^{cf}) \right) + (1 - y_i^{cf}) \log(1 - f(\mathbf{x}_i^{cf})) \right) \\
&= -\sum_{i=1}^{N^{tra}} \log \left( f(\mathbf{x}_i^{cf}) \right),
\end{aligned}
\tag{3}
$$

where $y_i^{cf}$ is the anticipated prediction value for CF. For binary classification tasks, the positive class probability is maximized by setting $y_i^{cf} = 1$.

Considering the transformation of continuous spectrum, we use the weighted square error to quantify proximity, which includes a hyperparameter $\boldsymbol{w}$ to adjust the perturbability of the features.

$$
\mathcal{L}_{wprox}(\boldsymbol{\theta}, \mathcal{D}) = \sum_{i=1}^{N^{tra}} \| \boldsymbol{w} * (\mathbf{x}_i^{tra} - \mathbf{x}_i^{cf}) \|_2^2,
\tag{4}
$$

where $*$ is the element product of the vectors.

**Optimization.** The overall optimization problem can be represented as

$$
\boldsymbol{\theta}^* = \underset{\boldsymbol{\theta}}{\operatorname{argmin}} \ \lambda \mathcal{L}_{nll}(\boldsymbol{\theta}, \mathcal{D}) + \mathcal{L}_y(\boldsymbol{\theta}, \mathcal{D}) + \mathcal{L}_{prox}(\boldsymbol{\theta}, \mathcal{D}).
\tag{5}
$$

In this equation, the hyperparameter $\lambda$ balances the importance of NLL, validity, and proximity.

Once FastDCFlow's parameters $\boldsymbol{\theta}$ are trained, the model can be fixed using these parameters. This approach accelerates the generation of a CF set for any test input $\mathbf{x}^{tes}$, thereby rendering it more efficient than input-based methods. To adjust diversity during generation without adding any explicit diversity loss term, we introduced a temperature parameter $t$ and added noise to the latent space with $\boldsymbol{\epsilon}_t \sim \mathcal{N}(\mathbf{0}, t\boldsymbol{I})$. Note that too high a temperature may reduce proximity and effectiveness performance.

Given $N^{tes}$ as the number of test inputs, with $\mathcal{T} = \{\mathbf{x}_i^{tes}\}_{i=1}^{N^{tes}}$ representing each input pair, and $M$ as the number of CFs generated for each input, this can be conducted in parallel. a comprehensive procedure, including training, is presented in Algorithm 1.

In TE transformation, two functions are utilized: fit_transformTE($\bullet$), which substitutes categorical variables in the training data with target means via out-of-fold calculations, and transformTE($\bullet$), which is the same for all training data. The CF sets derived from the test inputs were reverted to their initial categorical variables using the inverse_transformTE($\bullet$) function. During this step, a binary search tree assisted in mapping the variables to the closest categorical variable level.

## 4 EVALUATION

**Validation of TE effectiveness.** We employed two distinct training datasets, one using OHE and the other using TE, to train both the ML model for probability estimation and FastDCFlow. By evaluating the classification performance difference on the test data using a t-test, we ensured consistent performance across both encoding techniques (OHE and TE).

We examined the CE trends produced by FastDCFlow. For the $i$-th test input, let the predicted value be denoted as $\hat{y}_i^{tes} = f(\mathbf{x}_i^{tes})$ and the average predicted value across all test inputs as $\hat{y}^{tes} = \frac{1}{N^{tes}} \sum_i \hat{y}_i^{tes}$.

For each test input, a set of $M$ CFs was generated and represented as $\mathcal{S}_i = \{\mathbf{x}_{ij}^{cf}\}_{j=1}^{M}$. The $j$-th predicted value within this set was $\hat{y}_{ij}^{cf} = f(\mathbf{x}_{ij}^{cf})$. The average predicted value across these $M$

Table 1: Evaluation metrics. $I(\bullet)$ is the indicator function.

| Metrics | Symbol | Definition |
|---|---|---|
| Proximity | P | $P = \frac{1}{N^{tes}} \sum_i \frac{1}{M} \sum_j \|\mathbf{x}_i^{tes} - \mathbf{x}_{ij}^{cf}\|_2$ |
| Validity | V | $V = \frac{1}{N^{tes}} \sum_i \frac{1}{M} \sum_j I\left(f(\mathbf{x}_{ij}^{cf}) - f(\mathbf{x}_i^{tes}) > 0\right)$ |
| Inner diversity | ID | $ID = \frac{1}{N^{tes}} \sum_i \frac{1}{M(M-1)} \sum_{j \neq k} \|\mathbf{x}_{ij}^{cf} - \mathbf{x}_{ik}^{cf}\|_2$ |
| Outer diversity | OD | $OD = \frac{1}{N^{tes}(N^{tes}-1)} \sum_{i \neq k} \frac{1}{M^2} \sum_j \sum_l \|\mathbf{x}_{ij}^{cf} - \mathbf{x}_{kl}^{cf}\|_2$ |
| Run time | RT | $RT = \frac{1}{N^{tes}} \left(T_1(\boldsymbol{g_\theta}) + T_2(N^{tes}, M)\right)$ |

---

**Algorithm 1:** Generating CFs

---

**Data:** Training inputs $\mathcal{D} = \{\mathbf{x}_i^{tra}\}_{i=1}^{N^{tra}}$, test inputs $\mathcal{T} = \{\mathbf{x}_i^{tes}\}_{i=1}^{N^{tes}}$, pretrained binary
probability prediction model $f(\bullet)$.

1 Apply TE to training and test inputs to convert categorical variables into continuous variables:

$$\mathcal{D} \leftarrow \text{fit\_transformTE}(\mathcal{D}),$$
$$\mathcal{T} \leftarrow \text{transformTE}(\mathcal{T}).$$

2 Learn the parameters $\boldsymbol{\theta}^*$ of the FastDCFlow function $\boldsymbol{g_\theta}$ from NLL, validity, and proximity
losses:

$$\boldsymbol{\theta}^* = \underset{\boldsymbol{\theta}}{\arg\min} \ \lambda \mathcal{L}_{nll}(\boldsymbol{\theta}, \mathcal{D}) + \mathcal{L}_y(\boldsymbol{\theta}, \mathcal{D}) + \mathcal{L}_{wprox}(\boldsymbol{\theta}, \mathcal{D}).$$

3 **for** $i = 1, \cdots, N^{tes}$ **do**
4     Obtain latent variable $\mathbf{z}_i$ using forward transformation, $\mathbf{z}_i = \boldsymbol{g_\theta}(\mathbf{x}_i^{tes})$, $\mathbf{z}_i \sim p_{\mathcal{Z}}$.
5     **for** $j = 1, \cdots, M$ **do**
6         Obtain perturbed latent variable $\mathbf{z}_{ij}^* = \mathbf{z}_i + \boldsymbol{\epsilon}_{tj}$ with $\boldsymbol{\epsilon}_{tj} \sim \mathcal{N}(\mathbf{0}, t\boldsymbol{I})$.
7         Generate counterfactual sample $\mathbf{x}_{ij}^{cf} = \boldsymbol{g}_\theta^{-1}(\mathbf{z}_{ij}^*)$, $\mathbf{x}_{ij}^{cf} \sim p_{\mathcal{X}}$.
    **end**
8     Reverse-transform categorical variables to their original levels:

$$\mathbf{x}_{ij}^{cf} \leftarrow \text{inverse\_transformTE}(\mathbf{x}_{ij}^{cf}).$$

**end**

**Result:** Set of CFs $\{\mathcal{S}_i\}_{i=1}^{N^{tes}}$, where $\mathcal{S}_i = \{\mathbf{x}_{ij}^{cf}\}_{j=1}^M$.

---

CFs was $\hat{y}_i^{cf} = \frac{1}{M} \sum_j \hat{y}_{ij}^{cf}$, and the overall average predicted value for all CFs is expressed as
$\hat{y}^{cf} = \frac{1}{N^{tes}} \sum_i \hat{y}_i^{cf}$.

These metrics aid in ascertaining the changes in the CF prediction values. To address instances
wherein the CF prediction values were consistently the same irrespective of the test input, we also
validated them based on the standard deviation of $\hat{y}_i^{cf}$. Considering that OHE and TE yield varying
feature dimensions post-transformation, both the variability of $\hat{y}_i^{cf}$ for individual test inputs and the
standard deviation values were compared.

**Performance metrics for CE.** We introduced five specialized metrics to evaluate the CE and two
main metrics to capture their combined effects. Both FastDCFlow and its competitors employed
TE-transformed datasets, treating all features as continuous variables. Table 1 lists the metrics.

- **Proximity (P)**: Utilizing the $l_2$ norm, we measured how close each generated CF was to its
corresponding test input. Lower values are preferable.

- **Validity (V)**: We computed the mean improvement in the CFs' predicted outcomes $(y_{ij}^{cf})$
over the test inputs $(y_i^{tes})$. Higher values signify better performance.

- **Inner diversity (ID)**: This is measured as the $l_2$ distance between each CF pair within a
set of $M$ CFs for a single test input. Greater values indicate more internal diversity and are
deemed better.

- **Outer diversity (OD)**: This is the $l_2$ distance between CF pairs generated for different test inputs. A larger value signifies better performance through more diverse CFs.
- **Run time (RT)**: This metric considers two time components:
  - $T_1(g_\theta)$: This is the time for pretraining the CF generation model.
  - $T_2(N^{tes}, M)$: This is the time to generate $M$ CFs for each $N^{tes}$ test input.

## 5 EXPERIMENT

### 5.1 DATASETS AND PREPROCESSING

To assess the proposed approach, we leveraged three open datasets: Adult (Becker & Kohavi, 1996), Bank (Moro et al., 2012), and Churn (Macko, 2019), which integrate both categorical and continuous variables. The Adult dataset, sourced from the US Census, focuses on factors influencing an individual's annual income, classifying whether it exceeds $50k. The Bank dataset, which is from a Portuguese bank's marketing efforts, centers on whether customers opt for a term deposit. The Churn dataset from IBM presents fictional telecom customer churn indicators. Although the Adult dataset is distinct in its six categorical variables with a vast range of values, both the Bank and Churn datasets stand out for their high dimensionality, with the latter heavily leaning towards categorical data. Across all datasets, the task of the ML model was to predict the probability of a target variable. Dataset specifics can be found in the Appendix E, Table 6.

In our analysis, we prepared two versions of the datasets, transforming categorical variables using OHE and TE, and standardizing them to have a mean of 0 and variance of 1. The data is partitioned into a 90% training set and a 10% test set. The test inputs for CF generation are those with a predicted probability of less than 0.5 from the pretrained ML model. We implemented a 10-fold on the training data for TE.

### 5.2 BASELINES

To assess the effectiveness of TE, we employed datasets preprocessed with OHE and TE, which were evaluated using standard deviations. Compared with other competing models, six additional models were considered: DiCE (Mothilal et al., 2020), GeCo (Schleich et al., 2021), MOC (Dandl et al., 2020), CF_VAE (Mahajan et al., 2019), CF_CVAE, and CF_DUVAE. DiCE, GeCo, and MOC are input-based methods for generating counterfactuals, whereas the other methods employ model-based methods. CF_VAE uses VAE to generate counterfactuals. By contrast, CF_CVAE extends CF_VAE by incorporating a conditional variational autoencoder (CVAE) (Sohn et al., 2015) to handle category labels explicitly. Furthermore, CF_DUVAE extends CF_VAE by incorporating the DUVAE (Shen et al., 2021), which improves the diversity of the latent space by applying batch normalization (BN) (Ioffe & Szegedy, 2015) and dropout (Srivastava et al., 2014) to the encoder output of VAE. Similarly to FastDCFlow, a temperature parameter is introduced to VAEs for test inputs. Details of the objective functions and model architectures are provided in the Appendix C.

**Parameter setting.** To ensure a fair assessment, all models applied a consistent ML model based on their encoding methods and the weight of proximity loss is $w = 1$. DiCE, which does not require pretraining, was set for 100 iterations. Model-based methods were subjected to 10 pretraining epochs with a batch size of 64 and assumed a mixed Gaussian distribution (Izmailov et al., 2020; Duong et al., 2023), utilizing RealNVP (Dinh et al., 2016) for invertible transformations. The model details are provided in the Appendix C. For the GA-based GeCo and MOC, 100 generations with 1,000 individuals each were set. GeCo randomly generated initial entities, with the next generation comprising the top 100 and 900 offspring produced from a uniform crossover (40% from Parent 1, 40% from Parent 2, and a 20% mutation rate). MOC employed the NSGA2 algorithm (Deb et al., 2002), initiating entities evenly across dimensions using Latin hypercube sampling (LHS) (McKay et al., 2000) and populating subsequent generations through uniform crossover with a 50% crossover rate and a polynomial mutation of the distribution index 20. To determine the effectiveness of TE using FastDCFlow, $N^{tes} = 500$ with $M = 1000$ were set. In the comparative experiments, we considered the computational cost of the input-based method, $N^{tes} = 100$ with $M = 100$ CFs. The temperature parameter $t$ was fixed at 1.0 in FastDCFlow and VAEs, and the hyperparameter $\lambda$ was determined using a grid search set to 0.01.

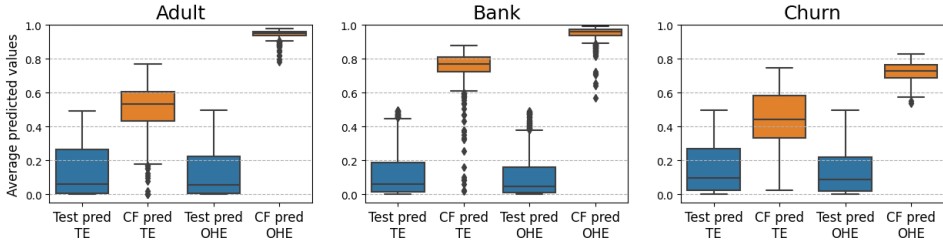

Figure 2: The boxplot showcases $\hat{y}_i^{tes}$ and $\hat{y}_i^{cf}$ values from FastDCFlow trained using TE and OHE across the Adult, Bank, and Churn datasets. In each graph, four boxplots, from left to right, represent: $\hat{y}_i^{tes}$ with TE (blue), $\hat{y}_i^{cf}$ with TE (orange), $\hat{y}_i^{tes}$ with OHE (blue), and $\hat{y}_i^{cf}$ with OHE (orange).

## 6 RESULTS AND ANALYSIS

### 6.1 EFFECTIVENESS OF TE

Figure 2 shows the difference in the predicted values between each test input and its counterfactuals, and the evaluation results are presented in Table 2. The $p$-val represents the significance of the difference in the predicted test input values obtained from the models trained on both TE and OHE. This shows that there is no significant distinction in $\hat{y}_i^{tes}$. Therefore, we focus on the CFs predicted values.

For every dataset, the OHE model consistently improved the predicted target values. For the Adult and Bank datasets, $\hat{y}_i^{cf}$ had an the upper limit of 1.0, whereas Churn exibited better results with OHE. However, the standard deviation values for OHE were smaller across datasets compared to those for TE. This suggests that OHE consistently generated counterfactuals irrespective of test input features be-

Table 2: Evaluation between OHE and TE.

| Dataset | | $p$-val | $\hat{y}^{tes}$ | $\hat{y}^{cf}$ |
|---|---|---|---|---|
| Adult | OHE | > 0.1 | 0.13 | $0.95 \pm 0.02$ |
| | TE | | 0.14 | $0.51 \pm \mathbf{0.14}$ |
| Bank | OHE | > 0.1 | 0.11 | $0.95 \pm 0.05$ |
| | TE | | 0.13 | $0.75 \pm \mathbf{0.11}$ |
| Churn | OHE | > 0.05 | 0.14 | $0.72 \pm 0.05$ |
| | TE | | 0.16 | $0.45 \pm \mathbf{0.16}$ |

cause categorical variables have a uniform perturbation cost in OHE. In contrast, TE, which transforms categorical variables into continuous ones, considered varying costs and better reflected the traits of each test input, generating more diverse predicted values.

### 6.2 OVERALL PERFORMANCE

Table 3 presents the evaluation metrics with the trade-offs. Consequently, our goal is not to develop a model that achieves the highest performance across all metrics. Rather, we demonstrate that FastDCFlow, a model-based method, offers comparable performance to existing Input-based methods with significantly reduced computational time. Focusing on the RT, model-based methods such as CF_VAE, CF_CVAE, CF_DUVAE, and FastDCFlow were faster than input-based methods such as DiCE, GeCo, and MOC. Thus, even with pretraining data time, latent variable optimization was more efficient in generating CFs. Although the V metrics were above 70% for all datasets and models, differences in performance emerged when evaluating metrics such as P, ID, and OD.

Despite the fact that a trade-off between diversity and proximity does not always hold, the P metric of DiCE underperformed, whereas its ID and OD metrics excelled. This is because of its integrated diversity loss term, which drives the creation of distinct CFs but may sacrifice proximity and predictive quality.

In contrast, GeCo exhibited superior P metric performance. However, its lower ID metric suggested that CFs from the same test input resembled each other, potentially because GeCo does not prioritize individual diversity in its generational mechanics. However, its strong OD metric indicated its adaptability to changing inputs. MOC improved GeCo's ID performance by using Pareto front estimation; however, it blended CFs focusing on predictive enhancement and those emphasizing proximity, resulting in a slightly diminished P metric.

Table 3: Evaluation results compared with competing models. For P and RT, lower values indicate better performance, while for other metrics, higher values signify superior performance. The best is in bold, the second-best is underlined, and the worst is marked with an asterisk.

| Dataset and model | | | P | V | ID | OD | RT |
|---|---|---|---|---|---|---|---|
| Adult | Input-based | DiCE | $4.7_{\pm0.70}{}^*$ | $0.82_{\pm0.21}{}^*$ | $\mathbf{2.2_{\pm0.51}}$ | $\mathbf{2.3_{\pm0.31}}$ | $\underline{15}$s |
| | | GeCo | $\mathbf{0.74_{\pm0.39}}$ | $\mathbf{1.0_{\pm0.00}}$ | $0.063_{\pm0.01}{}^*$ | $1.6_{\pm0.01}$ | 25s |
| | | MOC | $2.9_{\pm0.39}$ | $\mathbf{1.0_{\pm0.01}}$ | $0.97_{\pm0.13}$ | $1.5_{\pm0.53}$ | $30^*$s |
| | Model-based | CF_VAE | $3.1_{\pm0.86}$ | $\mathbf{1.0_{\pm0.00}}$ | $0.15_{\pm0.00}$ | $0.15_{\pm0.01}$ | $< \mathbf{1.0}$s |
| | | CF_CVAE | $3.0_{\pm0.87}$ | $\mathbf{1.0_{\pm0.00}}$ | $0.16_{\pm0.01}$ | $0.16_{\pm0.00}$ | $< \mathbf{1.0}$s |
| | | CF_DUVAE | $3.1_{\pm0.87}$ | $\mathbf{1.0_{\pm0.00}}$ | $0.13_{\pm0.00}$ | $0.13_{\pm0.00}{}^*$ | $< \mathbf{1.0}$s |
| | | FastDCFlow | $\underline{2.4_{\pm0.50}}$ | $0.99_{\pm0.04}$ | $0.98_{\pm0.04}$ | $\underline{1.8_{\pm0.94}}$ | $< \mathbf{1.0}$s |
| Bank | Input-based | DiCE | $25_{\pm0.92}{}^*$ | $0.97_{\pm0.03}{}^*$ | $\mathbf{9.5_{\pm0.20}}$ | $\mathbf{9.5_{\pm0.28}}$ | $1.4\times10^2{}^*$s |
| | | GeCo | $\mathbf{2.0_{\pm0.16}}$ | $0.98_{\pm0.05}$ | $1.0_{\pm0.06}$ | $2.4_{\pm1.92}$ | 39s |
| | | MOC | $11_{\pm1.17}$ | $\mathbf{1.0_{\pm0.00}}$ | $\underline{5.2_{\pm0.67}}$ | $\underline{5.9_{\pm1.58}}$ | $\underline{29}$s |
| | Model-based | CF_VAE | $4.6_{\pm1.56}$ | $\mathbf{1.0_{\pm0.00}}$ | $0.22_{\pm0.00}{}^*$ | $0.21_{\pm0.01}{}^*$ | $< \mathbf{1.0}$s |
| | | CF_CVAE | $4.6_{\pm1.57}$ | $\mathbf{1.0_{\pm0.00}}$ | $0.22_{\pm0.01}{}^*$ | $0.22_{\pm0.01}$ | $< \mathbf{1.0}$s |
| | | CF_DUVAE | $4.6_{\pm1.55}$ | $\mathbf{1.0_{\pm0.00}}$ | $0.22_{\pm0.01}{}^*$ | $0.22_{\pm0.01}$ | $< \mathbf{1.0}$s |
| | | FastDCFlow | $\underline{2.8_{\pm0.20}}$ | $0.97_{\pm0.04}{}^*$ | $1.6_{\pm0.03}$ | $3.0_{\pm1.74}$ | $< \mathbf{1.0}$s |
| Churn | Input-based | DiCE | $5.7_{\pm0.35}{}^*$ | $0.78_{\pm0.29}{}^*$ | $\mathbf{2.3_{\pm0.03}}$ | $\underline{2.3_{\pm0.05}}$ | $1.4\times10^2{}^*$s |
| | | GeCo | $\mathbf{1.5_{\pm0.12}}$ | $0.90_{\pm0.14}$ | $0.75_{\pm0.04}$ | $2.6_{\pm1.2}$ | 43s |
| | | MOC | $4.1_{\pm0.49}$ | $\mathbf{1.0_{\pm0.00}}$ | $1.4_{\pm0.14}$ | $2.4_{\pm0.78}$ | $\underline{30}$s |
| | Model-based | CF_VAE | $5.3_{\pm1.33}$ | $0.86_{\pm0.30}$ | $0.26_{\pm0.00}$ | $0.26_{\pm0.01}$ | $< \mathbf{1.0}$s |
| | | CF_CVAE | $5.3_{\pm1.30}$ | $0.88_{\pm0.30}$ | $0.25_{\pm0.00}{}^*$ | $0.25_{\pm0.01}{}^*$ | $< \mathbf{1.0}$s |
| | | CF_DUVAE | $5.3_{\pm1.38}$ | $0.90_{\pm0.28}$ | $0.25_{\pm0.01}{}^*$ | $0.25_{\pm0.01}{}^*$ | $< \mathbf{1.0}$s |
| | | FastDCFlow | $\underline{3.6_{\pm0.73}}$ | $0.94_{\pm0.05}$ | $1.8_{\pm0.11}$ | $\mathbf{3.2_{\pm0.93}}$ | $< \mathbf{1.0}$s |

Models rooted in the VAE, including CF_VAE, CF_CVAE, and CF_DUVAE, consistently generated similar CFs irrespective of the test input variation. This be attributed to their inherent leanings towards minimizing the KL divergence over optimizing the predictive accuracy, which is particularly evident in CF_VAE. Whereas, although CF_DUVAE sought latent space diversity, it fell short of with data rich in categorical variables, suggesting that the VAEs Gaussian distribution assumption in the latent space might not be suitable for intricate tabular data. The details of the VAEs training process can be found in Appendix D.2.

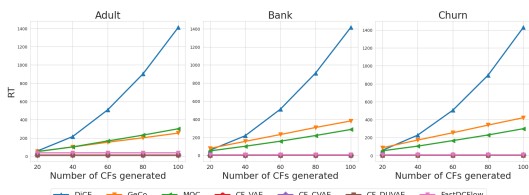

Figure 3: Relationship between $M$ and RT for each model.

FastDCFlow, while not leading in any individual metric, exibited a balanced high-performance when stacked against its peers. Specifically, it maintained a modest P value while bolstering the ID and OD metrics, as exemplified in its Adult and Churn dataset performance. This indicates FastDCFlow's ability to negotiate the intricacies of counterfactual generation, thereby side-stepping the challenges that VAEs face in tabular data diversity. Thus, it ensures that latent space alterations resulted in diverse original feature modifications. In Bank dataset, FastDCFlow showed a slightly lower performance than other input-based methods. especially for ID and OD, it is inferior to DiCE and MOC, which is because DiCE sacrifices predictive accuracy and proximity in order to emphasize diversity. On the other hand, MOC mixes CFs that emphasize predictive accuracy and CFs that emphasize proximity because it uses pareto front estimation, which results in a slight decrease in P. Overall, FastDCFlow, while being a model-based method, demonstrated the ability to generate more diverse Counterfactuals (CFs) than VAE, showing performance comparable to input-based methods. Additionally, it was shown to produce well-balanced CFs in a shorter computation time.

## 6.3 THE EFFECT OF CF PARAMETERS

**Sample size effect:** Figure 3 presents a comparison of the relationship between $M$ and RT ($N^{tes} = 10$ and $M = 20, 40, \cdots, 100$). Model-based methods can significantly reduce the RT. These models,

excluding pretraining time, generate CFs through latent space mapping and become more practical with increase in $M$. The RT of DiCE increases exponentially with the number of generations, owing to the computational complexity of $\mathcal{O}(KM^2 + M^3)$ required for the diversity loss term in datasets with dimension number $K$.

**Temperature effect:** Figure 4 shows the trends of P, V, ID, and OD when varying the temperature parameter of model-based methods ($N^{tes} = 100$ and $M = 100$). For Fast-DCFlow, increases in temperature correlated with increases in P, ID, and OD, indicating that counterfactuals learned the necessary constraints and considered proximity and diversity within the latent space. V is defined as the proportion of improved predictions for test inputs; therefore, it is possible to enhance diversity without deteriorating its value. In contrast, methods based on VAE showed no variation in the evaluation metrics regardless of temperature changes, suggesting that these models do not adapt to the characteristics of the test inputs. Furthermore, to investigate the relationship between DiCE's explicit diversity loss and FastD-CFlow temperature, the changes in the logdet value of the diversity loss (DiCE logdet, see Appendix C.1) are also shown in line 5. The results confirm that the logdet value increases with temperature changes and reaches a steady state when the temperature is sufficiently high.

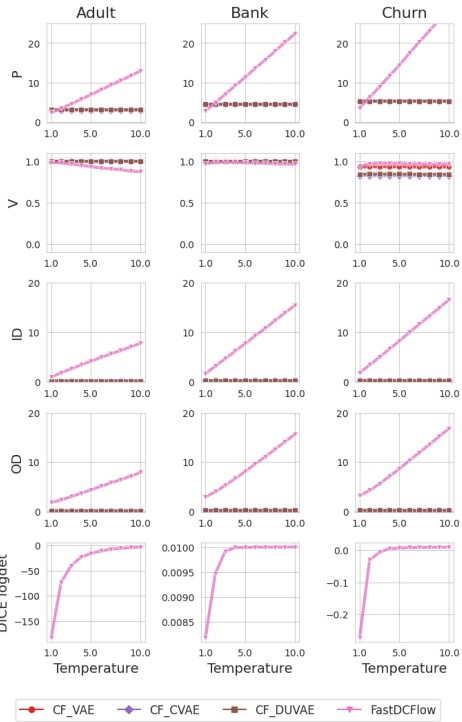

Figure 4: Changes in evaluation metrics with adjustments to the temperature parameter in model-based methods. Each line represents an evaluation metric (including DiCE diversity loss), and each column corresponds to a dataset.

## 7 CONCLUSION

This study applied a TE transformation that considered the perturbation of categorical variables in CE, thus tackling the problem of CF's predicted values tending to the upper limit. In addition, we introduced a method to learn the latent space in alignment with the CF constraints by utilizing a normalization flow termed FastDCFlow. This latent space, which was derived from the training data, offered efficient generation for any test input. Our experimental findings confirmed the proficient balance that approach maintained among the evaluation metrics, rendering it the best in terms of overall assessments. The superior performance of FastDCFlow was largely owing to its advancements in diversity and speed over input-based methods. Moreover, in contrast to VAE-based methods, FastDCFlow captured a precise depiction of the training data, enabling CF generation through the inverse transformation of proximate points.

A limitation of this work is fourfold. Firstly, when applying TE with certain categories that feature seldom seen levels or are data-deprived, there is a looming threat of overfitting, owing to stark fluctuations in the target variable. Secondly, given that decision-makers are human, it is impossible to ignore user feedback and biases originating from the domain. While FastDCFlow offers new possibilities for utilizing model-based methods, qualitative evaluation of the generated CEs and detection of biases remain as future work. Thirdly, as both input-based and model-based methods require ML training, sparse high-dimensional data can hinder the utility of CEs. Fourthly, the applicability of ML is limited to differentiable models. Although decision tree-based models are commonly used with tabular data, they are not applicable in FastDCFlow.

In subsequent phases of this study, a deeper exploration is imperative to understand the underlying mechanisms of how the model portrays intricate tabular data distributions.

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

Table 4: Example of a test input.

| Feature | Value |
|---|---|
| Age | 58 |
| Workclass | Self-Employed |
| Education | HS-grad |
| Marital Status | Married |
| Occupation | White-Collar |
| Race | White |
| Gender | Male |
| Hours per Week | 60 |
| Income (predicted value) | 0.24 |

## A    INTERPRETATION OF CFS

The categorical variables in the set of CFs generated by FastDCFlow can be interpreted to determine which ones should be perturbed, based on their predicted values. Since tabular data is composed of continuous and categorical variables, we employed the latent class regression model (LCRM) (Dayton & Macready, 1988; Hagenaars & McCutcheon, 2002). When applying the LCRM, it is assumed that there are latent classes underlying the generated categorical variables, and each CF is considered to be probabilistically classified into one of a classes. The probability of each class is regressed by continuous variables; thus, transitions in class probabilities can be estimated based on changes in the ML predicted values. For this estimation, multinomial logistic regression (Agresti, 2012) is used.

We used a record from the Adult dataset as the test input, detailed in Table 4. Post TE, the ML model predicted a value of 0.24. In our experiments, we generated $M = 1000$ CFs. By increasing the number of generated counterfactuals, we have the advantage of interpreting counterfactuals as a distribution. We apply the LCRM to the generated set of CFs, utilizing the poLCA R library (Linzer & Lewis, 2011). In the visualization, we fix the 'age' at 63 (input + 5 years) and "hours_per_week" at 60 (the same as the input), and vary the predicted value of "income" from 0.0 to 1.0. Figure 5a illustrates the transition of class membership probabilities. From observing the transitions in class probabilities, we can infer that class 3 has a high probability when the predicted value is improved up to approximately 0.4; class 1 is likely around a predicted value of 0.7; and beyond that point, class 2 becomes highly probable.

Figure 5b illustrates the categorical distribution for each class. From this, we infer that in class 1, adjusting the "workclass" to "private" and the "occupation" to "sales" is suitable. Similarly, in class 2, modifying "workclass", 'education', and "occupation" seems optimal, and in class 3, alterations to "workclass" and "occupation" are deemed appropriate. On the other hand, some features in real-world data, like race or gender, can be difficult to modify or perturb. For these types of features, applying stricter penalty coefficients to uphold proximity constraints could be a viable approach.

## B    APPLAYING DOMAIN CONSTRAINTS

Depending on the type of dataset and specific domain constraints aligned with the objectives, it is possible to apply counterfactual explanations appropriately. In reality, characteristics such as gender or race cannot be perturbed. Furthermore, age must always satisfy a monotonic increase, making explanations that fail to adhere to these constraints infeasible. This section discusses the configuration of mutable features and the monotonic constraints using FastDCFlow.

Initially, the perturbation constraints on features can be achieved by adjusting the weights $\boldsymbol{w}$ of the proximity loss.

$$\mathcal{L}_{wprox}(\boldsymbol{\theta}, \mathcal{D}) = \sum_{i=1}^{N^{tra}} \|\boldsymbol{w} * (\mathbf{x}_i^{tra} - \mathbf{x}_i^{cf})\|_2^2. \tag{6}$$

Specifically, increasing the weights for features that should not be perturbed effectively imposes stronger penalties during perturbation. For the monotonic constraints, the following hinge loss func-

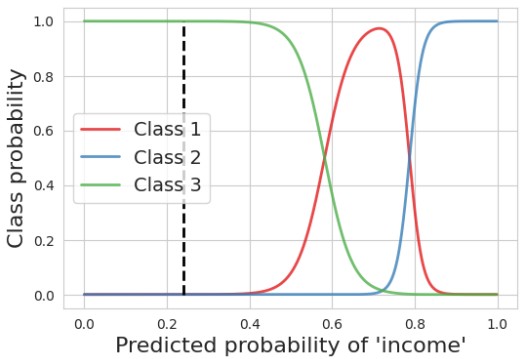

(a) The application results of LCRM on the set of CFs. The number of classes is determined where the bayesian information criterion (BIC) (Schwarz, 1978) is minimized, resulting in three classes. The horizontal axis represents the ML predicted values, and the vertical axis represents the class probabilities. The black dotted line represents the predicted value of the test input.

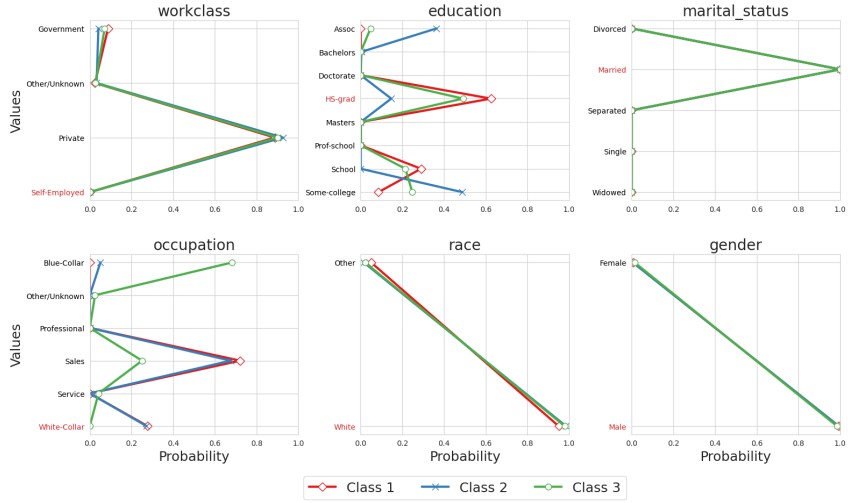

(b) Categorical distribution by the LCRM. Each graph represents a different categorical variable, with probability on the horizontal axis and categorical values on the vertical. Values in red denote input values, suggesting that categories with lower probabilities are more likely to be perturbed.

Figure 5: The application result of LCRM.

tion is added:

$$\mathcal{L}_{mon} = \frac{1}{|\mathcal{D}^{mon}|N^{tra}} \sum_{i=1}^{N^{tra}} \sum_{d \in D^{mon}} \max(x_{id}^{tra} - x_{id}^{cf}, 0), \tag{7}$$

where $D^{mon}$ is the set of features that should be monotonically increased.

**Experiment.** In the context of the Adult dataset, we compare the performance before and after the addition of domain constraints. For the evaluation, we apply the following two metrics specifically to the features with imposed constraints, and also use P, V, ID, and OD used in the Table 3 for the overall evaluation.

**Fix accuracy (FA):**

$$FA = \frac{1}{N^{tes}} \sum_{i=1}^{N^{tes}} \frac{1}{|\mathcal{D}^{fix}|M} \sum_{j=1}^{M} \sum_{d \in \mathcal{D}^{fix}} I\left(x_{ijd}^{cf} = x_{id}^{tes}\right), \tag{8}$$

Table 5: The results of domain addaptation.

| Method | P | V | FA | MA |
|---|---|---|---|---|
| No constraints | $3.6_{\pm 0.73}$ | $0.94_{\pm 0.05}$ | $0.97_{\pm 0.13}$ | $0.88_{\pm 0.11}$ |
| Domain constraints | $\mathbf{2.3}_{\pm \mathbf{0.50}}$ | $\mathbf{0.97}_{\pm \mathbf{0.10}}$ | $\mathbf{0.99}_{\pm \mathbf{0.11}}$ | $\mathbf{0.91}_{\pm \mathbf{0.09}}$ |

**Monotonicity accuracy (MA):**

$$\text{MA} = \frac{1}{N^{tes}} \sum_{i=1}^{N^{tes}} \frac{1}{|\mathcal{D}^{mon}|M} \sum_{j=1}^{M} \sum_{d \in \mathcal{D}^{mon}} I\left(x_{ijd}^{cf} > x_{id}^{tes}\right), \quad (9)$$

where $\mathcal{D}^{fix}$ is the sets of features with fixed constraints. FA and MA are metrics that evaluate the proportion of CFs that satisfy the fixed and monotonic constraints, respectively. These metrics are evaluated by returning the input data and CFs to their original scale and category before evaluation.

For the features that should remain fixed, "gender" and "race" are selected, with their respective weight coefficients set to 3.0, while setting the coefficients for all other features to 1.0. Additionally, "age" is designated as a feature that must satisfy the monotonic increase constraint. The results are shown in Table 5 for $N^{tes} = 500$ and $M = 1000$. The results show that the addition of domain constraints outperforms the model without constraints in all metrics. This suggests that the addition of domain constraints can be effective in generating CFs that are more appropriate for the domain. However, it is important to note that the addition of domain constraints can lead to a decrease in diversity. Therefore, it is necessary to consider the trade-off between diversity and domain constraints when applying domain constraints.

## C  DETAILED SETTINGS OF THE COMPARED MODELS

### C.1  OBJECTIVE FUNCTIONS

To ensure a fair comparison, we maintain consistency in the objective functions related to effectiveness and proximity across all models. The results, as depicted in Table 3, are derived from models trained with all categorical variables converted using TE. In contrast to comparison models, which calculate proximity separately for continuous and categorical variables, our experiment considers all variables as continuous, eliminating the need for such differentiation.

**DiCE.** This method directly defines CFs as parameters and simultaneously generates $M$ CFs for a single input $\mathbf{x}^{tes}$. When the set of generated CFs is denoted as $\mathcal{S}$, the overall optimization is given by the following equation:

$$\mathcal{S} = \underset{\mathbf{x}_1^{cf}, \cdots, \mathbf{x}_M^{cf}}{\arg \min} -\sum_{i=1}^{M} \log\left(f(\mathbf{x}_i^{cf})\right) + \alpha_1 \sum_{i=1}^{M} \|\boldsymbol{w} * (\mathbf{x}^{tes} - \mathbf{x}_i^{cf})\|_2^2 - \alpha_2 \det\left(\boldsymbol{A}\right), \quad (10)$$

where $A_{ij} = \frac{1}{1+\|\mathbf{x}_i^{cf} - \mathbf{x}_j^{cf}\|_2^2}$ and $\det\left(\boldsymbol{A}\right)$ represents the loss term indicating diversity within the CFs. $\alpha_1$, $\alpha_2$ are hyperparameters adjusting the importance of each term. In line 5 of Figure 4, the logarithm of the determinant, $\log \det\left(\boldsymbol{A}\right)$, is shown for each temperature.

**Methods using GAs.** For the comparative models GeCo and MOC, CFs are generated in a similar manner to minimize the objective function related to validity and proximity. Given each individual $\mathbf{x}^{cf}$ as an instance of a CF, the overall objective function $f_{\text{all}}$ is defined by the following equation:

$$f_{\text{all}} = -\log\left(f(\mathbf{x}^{cf})\right) + \alpha_3 \|\boldsymbol{w} * (\mathbf{x}^{tes} - \mathbf{x}^{cf})\|_2^2. \quad (11)$$

$\alpha_3$ is a hyperparameter to adjust the importance of each term. If we maximize the objective function, the signs on the right side of $f_{\text{all}}$ should be reversed. Unlike DiCE, no explicit diversity term is added to the objective function; instead, CFs are generated depending on the optimization algorithm.

**Methods using VAEs.** CF_VAE, CF_CVAE, and CF_DUVAE fall under model-based methods, and the maximization of the likelihood is approximated by maximizing the variational lower bound (Kingma & Welling, 2013). Due to this approximation, the proximity in the loss function corresponds to the data reconstruction loss, and a normalization term via KL divergence is added.

Suppose we have a training dataset, denoted as $\mathcal{D} = \{\mathbf{x}_i^{tra}\}_{i=1}^{N^{tra}}$. VAEs comprise an encoder and a decoder. The encoder, characterized by the parameter $\phi$, can be represented as $q_\phi(\mathbf{z}|\mathbf{x}^{tra})$, and it is responsible for mapping the training data to the latent variable $\mathbf{z}$. Conversely, the decoder, characterized by the parameter $\theta$, can be represented as $p_\theta(\mathbf{x}^{cf}|\mathbf{z})$, and it maps the latent variable $\mathbf{z}$ back to the data space, producing counterfactual examples $\mathbf{x}^{cf}$. The model is trained by optimizing an overall loss function, represented by the following equation:

$$\mathcal{L}(\phi, \theta, \mathcal{D}) = \mathbb{E}_{q_\phi(\mathbf{z}|\mathbf{x}^{tra})||p_\theta(\mathbf{z})}[\log p_\theta(\mathbf{x}^{cf}|\mathbf{z})]$$
$$- \alpha_4 D_{\mathrm{KL}}(q_\phi(\mathbf{z}|\mathbf{x}^{tra})||p_\theta(\mathbf{z})) + \alpha_5 \mathcal{L}_y(\phi, \theta, \mathcal{D}). \tag{12}$$

Here, $\alpha_4$, $\alpha_5$ are hyperparameters to adjust the importance of each term and $D_{\mathrm{KL}}(\bullet)$ represents the KL divergence. Assuming the prior distribution of the decoder $p_\theta(\mathbf{z})$ to be $\mathcal{N}(\mathbf{0}, \boldsymbol{I})$ and the conditional distribution of the encoder $q_\phi(\mathbf{z}|\mathbf{x}^{tra})$ to be $\mathcal{N}(\boldsymbol{\mu}, \boldsymbol{\sigma}^2)$, it can be determined as follows using monte carlo approximation:

$$\mathcal{L}(\phi, \theta, \mathcal{D}) = \sum_{i=1}^{N^{tra}} \|\mathbf{x}_i^{tra} - \mathbf{x}_i^{cf}\|_2^2$$
$$+ \frac{\alpha_4}{2} \sum_{i=1}^{N^{tra}} (1 + \log(\sigma_j^2) - \mu_j^2 - \sigma_j^2) - \alpha_5 \sum_{i=1}^{N^{tra}} \log(f(\mathbf{x}_i^{cf})). \tag{13}$$

The hyperparameters $\alpha_1$ to $\alpha_5$ are fixed at 1.0 in the experiments.

## C.2 MODEL ARCHITECTURES

The ML model for predicting the target variable is constructed using a three-layer binary classification neural network. ReLU activation functions and dropout (with 0.5) are applied between each layer. The final output is obtained as a probability in the range $[0, 1]$ from a fully connected layer with 64 units, passed through a sigmoid function. The gradient descent optimization algorithm used is adam (Kingma & Ba, 2014), with a fixed learning rate of $10^{-3}$. The encoders for CF_VAE, CF_CVAE, and CF_DUVAE are multi-layer neural networks that incorporate BN, dropout (with 0.1), and the ReLU. VAE and DUVAE have three layers, while CVAE, due to the added dimensionality from labeling categorical variables, is constructed with four layers. For CF_DUVAE, when mapping to the latent space, BN is applied to the mean, and dropout (with 0.5) is applied to the variance. Regarding the RealNVP used in FastDCFlow, there are 3 coupling layers, and each coupling layer consists of 6 intermediate layers containing ReLU and dropout.

All codes are executed in Python 3.9, equipped with the PyTorch library and the pymoo library (for MOC) (Blank & Deb, 2020). The machine specifications used for the experiments are Debian GNU/Linux 11 (bullseye), Intel Core i9 3.30GHz CPU, and 128GB of memory.

## D TRANSITION OF OBJECTIVE FUNCTIONS

### D.1 INPUT-BASED METHODS

Figure 6 illustrates the progression of the objective function for specific test inputs in DiCE, GeCo, and MOC. Since input-based methods do not require training data, the objective function is optimized for each input. Observing DiCE in Figure 6a, all terms of the objective function seem to decrease linearly, suggesting that the learning is progressing smoothly. However, the rate of change in the objective function varies significantly depending on the dataset, and it is believed to be related to the dimensionality of the input and the number of CFs generated.

For GoCo and MOC, utilizing GA, shown in Figure 6b, 6c, the objective function demonstrates similar convergence across datasets. In both cases, improvement in proximity tends to be prioritized, and there is variation in the values of validity; for datasets other than Adult, it deteriorates as the number of generations increases. However, since the total loss is on a declining trend, and considering the model performance comparison in Table 3, it outperforms DiCE and VAE-based methods, it can be said that learning is progressing with consideration of the balance in trade-offs. Figure 7 represents the Pareto front of the top 100 CFs obtained in the final generation in MOC. It is evident that, in every dataset, MOC generates CF considering diversity.

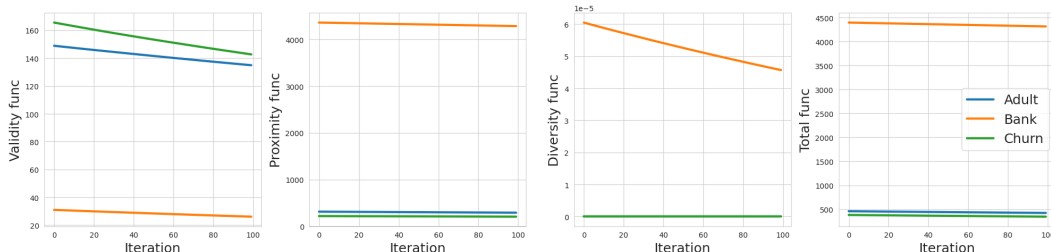

(a) Epoch-wise transition of the objective function in DiCE. From left to right, they correspond to the functions of validity, proximity, diversity, and total.

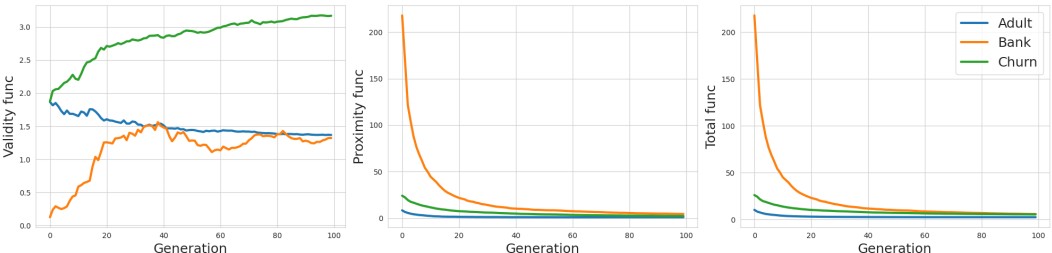

(b) Epoch-wise transition of the objective function in GeCo. From left to right, they correspond to the functions of validity, proximity, and total.

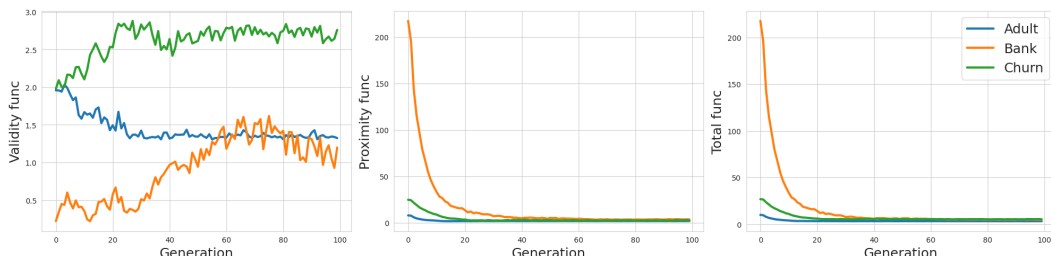

(c) Epoch-wise transition of the objective function in MOC. From left to right, they correspond to the functions of validity, proximity, and total.

Figure 6: Transition of objective function in input-based methods. The colors in the graph represent different types of datasets.

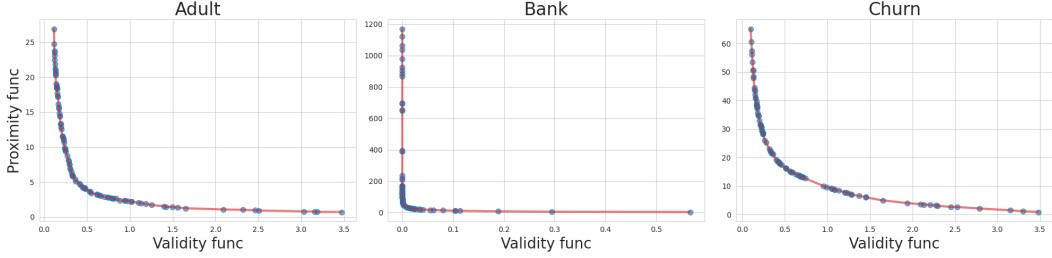

Figure 7: The Pareto front of the top 100 individuals in the final generation in MOC.

## D.2 MODEL-BASED METHODS

Figure 8 displays how the loss function evolves with each epoch for the model-based methods. In FastDCFlow, there is a steady decline in NLL. When considering the trade-off between validity and proximity, it is seen that as validity improves with each epoch, proximity seems to worsen, especially noticeable in the Adult and Churn datasets. However, due to the overall decreasing trend

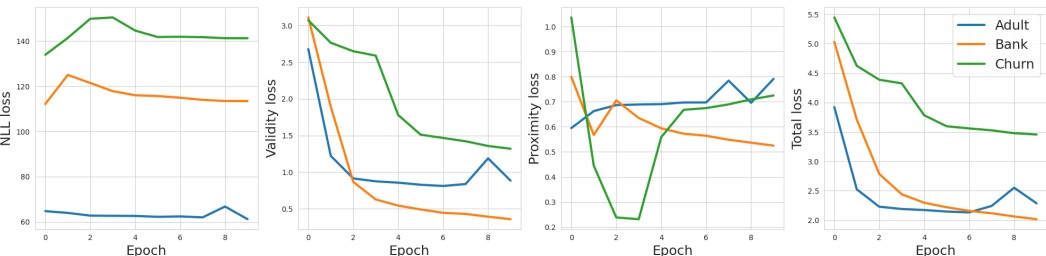

(a) Epoch-wise transition of the loss function in FastDCFlow. From left to right, they correspond to the losses of NLL, validity, proximity, and total.

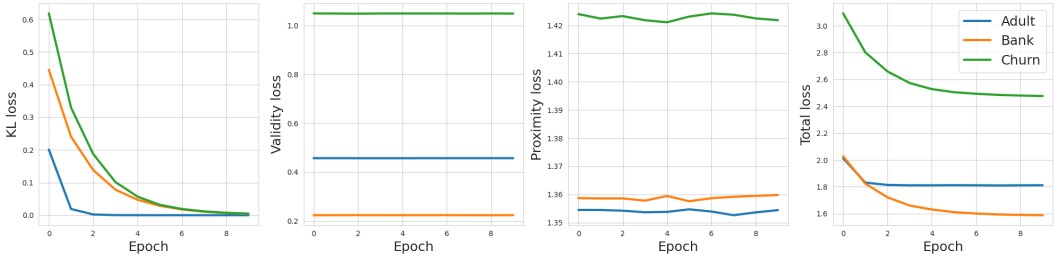

(b) Epoch-wise transition of the loss function in CF_VAE. From left to right, they correspond to the losses of KL divergence, validity, proximity, and total.

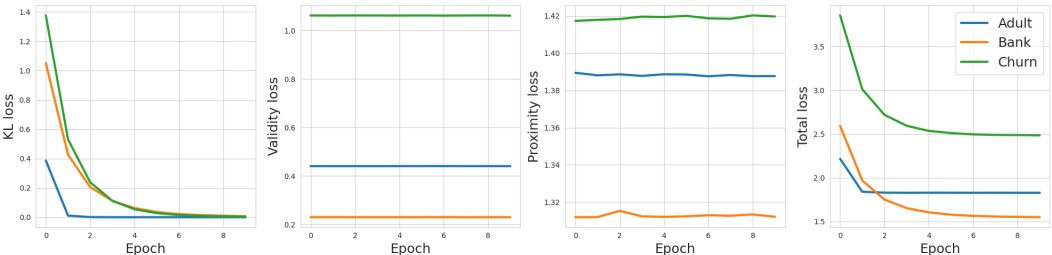

(c) Epoch-wise transition of the loss function in CF_CVAE. From left to right, they correspond to the losses of KL divergence, validity, proximity, and total.

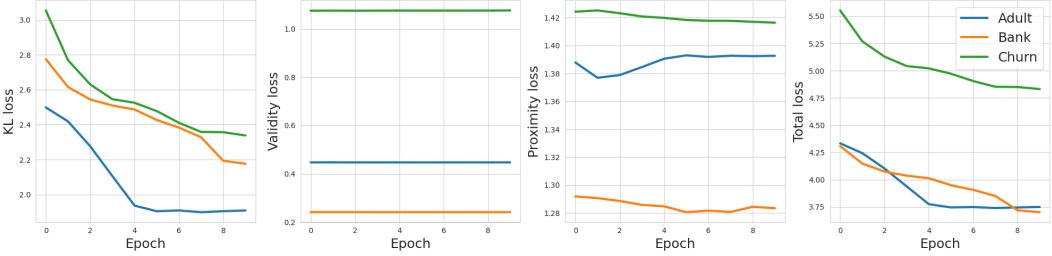

(d) Epoch-wise transition of the loss function in CF_DUVAE. From left to right, they correspond to the losses of KL divergence, validity, proximity, and total.

Figure 8: Transition of loss function in model-based methods. The colors in the graph represent different types of datasets.

in total loss, it is inferred that the model is giving precedence to learning validity over maintaining proximity.

For CF_VAE and CF_CVAE, Figure 8b, 8c show that the loss of KL is prioritized for improvement, with proximity showing slight variability in values. On the other hand, almost no improvement is observed in validity throughout the epochs, revealing that it is not contributing to the total loss. This issue is known as the phenomenon of posterior collapse (Zhao et al., 2019), where the posterior

Table 6: Dataset composition and predictive accuracy of ML models.

| Dataset | | # Rec | # Cat | # Con | # Dim | ACC | $p$-val | AUC | $p$-val |
|---------|-----|--------|-------|-------|-------|------------------|---------|------------------|---------|
| Adult | OHE | 32,561 | 6 | 2 | 30 | $0.83 \pm 0.0029$ | $> 0.05$ | $0.89 \pm 0.0050$ | $> 0.1$ |
|         | TE  |        |   |   | 8 | $0.83 \pm 0.0022$ |         | $0.88 \pm 0.0049$ |         |
| Bank | OHE | 11,162 | 9 | 7 | 52 | $0.85 \pm 0.0038$ | $> 0.01$ | $0.92 \pm 0.0051$ | $> 0.01$ |
|      | TE  |        |   |   | 16 | $0.84 \pm 0.0051$ |         | $0.91 \pm 0.0048$ |         |
| Churn | OHE | 7,043 | 16 | 3 | 47 | $0.80 \pm 0.012$ | $> 0.1$ | $0.85 \pm 0.074$ | $> 0.1$ |
|       | TE  |        |    |   | 19 | $0.80 \pm 0.012$ |         | $0.84 \pm 0.012$ |         |

distribution of the latent variables matches the prior distribution, leading to an inability to capture the characteristics of the data. As the match with the prior distribution is prioritized, it is evident that the loss of KL converges near zero in all datasets. The evaluation experiments illustrated in the Table 3 and Figure 4 indicate that the methods using VAE lacked diversity, and it can be said that a latent representation which properly considers the information for each input could not be obtained.

CF_DUVAE is a method that reduces the issue of posterior collapse in VAEs without requiring additional training or modifications to the model. In this model, by applying BN and dropout to the parameters of the posterior distribution, the preferential improvement of KL is restrained, enhancing the diversity of the latent variables. Observing the transition of the loss function in the Figure 8d, it is confirmable that the value of KL loss is restrained compared to other VAE-based methods. Also, the proximity is seen to decrease gently in the Bank and Churn datasets. However, like others, no improvement is observed in validity. As the Table 3 and Figure 4 illustrate, although the total loss is decreasing, the counterfactual diversity for the test inputs did not improve. Therefore, the effectiveness of DUVAE for counterfactual explanations using tabular data could not be confirmed.

# E DETAILS OF DATASETS

Table 6 displays the details of the datasets used in the experiments and the classification performance of ML models for each encoding method. # Rec, # Cat, # Con, and # Dim represent the number of records, count of categorical variables, count of continuous variables, and the dimension after encoding. While Adult has the highest number of records, its # Dim is the smallest. Bank has many levels for its categorical variables, making its # Dim the largest with OHE. Churn has the most categorical variables, but tends to have fewer values compared to the others.

Based on a 5-fold cross-validation on each dataset, models using OHE tend to have slightly better predictive accuracy. However, according to the t-test results, there's no significant performance difference except in the case of Bank, where there is a performance gap of about 0.01. However, given that the dimension of OHE is more than twice that of TE, considering the computational cost for CF generation, it can be argued that there is ample justification to apply TE.

