# OpenReview forum: "FastDCFlow: Fast and Diverse Counterfactual Explanations Using Normalizing Flows"
_ICLR.cc/2024/Conference — Submitted to ICLR 2024_

### Official Review · Reviewer_uvnA · 2023-10-25

**Soundness:** 3 good
**Presentation:** 3 good
**Contribution:** 2 fair
**Rating:** 5
**Confidence:** 4

**Summary:**

The paper proposes a methodology for computing counterfactual explanations using normalizing flows.

**Strengths:**

- Very interesting and novel idea of using normalizing flows
- Paper is well structured and mostly well written -- see Section "Weaknesses" for some criticism

**Weaknesses:**

- Experimental results do not look that convincing to me -- often other methods outperform the proposed method. Not sure how much sense the corrected scores make -- I think it would be better to compare different aspects directly, instead of merging them all into a single score
- Training a generation mechanism (e.g. the proposed method) might not be possible or difficult if only very few training samples are available. This might hinder the use of the proposed method in case of high-dimensional data space where only few samples are available
- Access to model internals are needed -- otherwise a gradient-based optimization method can not be applied to the proposed loss function for training the model. To me this looks as a another limitation of the proposed method
-  In general, I miss a discussion of limitations and drawbacks of the proposed method

**Questions:**

-  I am not sure how fair/appropriate the runtime comparison is: Other methods (non-generative methods) where never designed to be trained on data, so it is somewhat clear that these will not be able to compete with a trained generative method. On the other hand, building/training a generative method requires a lot of training data which poses a major disadvantage compared to other non-generative methods. I think a comparison of these very different methods is challenging -- maybe the authors can elaborate on this a bit more.

---

> ### Author Response · Authors · 2023-11-23
> **Thank you for your valuable review**
>
> Dear Reviewer uvnA,
>
> Thank you for your valuable feedback and suggestions on improving our paper. Here we address responses to your questions.
>
> At the beginning, we summarize significant revisions to this paper.
>
> **1) Modification in Implementation**
>
> Upon reviewing our source code, we discovered that for all model-based methods (CF_VAE, CF_CVAE, CF_DUVAE, FastDCFlow), the training data used was limited to samples where the ML prediction was < 0.5. However, the correct approach is to utilize the entire training dataset regardless of the predicted values. We have corrected this in our source code, retrained these models, and updated the results in our paper accordingly.
>
> **2) Revision of Experimental Goals**
>
> We have revised the manuscript to reflect our aim of developing a model that performs comparably to Input-based methods, with reduced computational time. Our objective is not to develop a model that excels in every evaluation metric but to create a model that is faster than Input-based methods while still maintaining balanced high performance across all metrics. While Input-based methods have the advantage of learning from each input and hence higher representational power, they are also time-consuming. We have incorporated the results of our revised experiments and discussed our findings in relation to the paper's objectives.

---

> > ### Author Response · Authors · 2023-11-23
> > **Thank you for your valuable review**
> >
> > **Q1.** Experimental results do not look that convincing to me -- often other methods outperform the proposed method. Not sure how much sense the corrected scores make -- I think it would be better to compare different aspects directly, instead of merging them all into a single score.
> >
> > **A1.** As initially mentioned, there was a training error in our data, which led us to re-conduct the relevant experiments and reflect the results accordingly. Furthermore, as you rightly pointed out, the overall performance evaluation has been a point of contention for us. In our paper, we defined combined metrics, CV and CS, but their arbitrariness in setting posed a problem. Acknowledging this, we have decided to discontinue the performance comparison based on CV and CS and instead focus solely on individual evaluation metrics. This revision allows for a more direct and transparent comparison of different aspects of performance.
> >
> > **Q2.** Training a generation mechanism (e.g. the proposed method) might not be possible or difficult if only very few training samples are available. This might hinder the use of the proposed method in case of high-dimensional data space where only few samples are available.
> >
> > **A2.** Your point is indeed valid. However, the limitation of having only a few training samples is a serious concern that affects both model-based and input-based methods equally. This is because both approaches require training data for learning the probability predictor (ML model f), making it a general issue rather than one specific to model-based methods. Accordingly, we have addressed this limitation in Chapter 7, Conclusion, under the limitations section to acknowledge this critical aspect of our methodology.
> >
> > **Q3.** Access to model internals are needed -- otherwise a gradient-based optimization method can not be applied to the proposed loss function for training the model. To me this looks as a another limitation of the proposed method.
> >
> > **A3.** Your observation is indeed accurate. It is a limitation of our method that models like decision trees, which are not differentiable, cannot be applied. We have acknowledged this limitation in the Conclusion (Chapter 7) of our paper, highlighting the constraint regarding the need for access to model internals for the application of gradient-based optimization methods in training our model.
> >
> > **Q4.** In general, I miss a discussion of limitations and drawbacks of the proposed method.
> >
> > **A4.** You are correct in noting that a clearer articulation of the model's limitations was needed. To address this, we have compiled a detailed discussion of the limitations in Chapter 7, under the Conclusion section of our paper. This includes an overview of the drawbacks and challenges associated with the proposed method.
> >
> > **Q5.** I am not sure how fair/appropriate the runtime comparison is: Other methods (non-generative methods) where never designed to be trained on data, so it is somewhat clear that these will not be able to compete with a trained generative method. On the other hand, building/training a generative method requires a lot of training data which poses a major disadvantage compared to other non-generative methods.
> >
> > **A5.** It's not always accurate to say that model-based methods (generative methods) are superior to input-based methods (non-generative methods). Firstly, both approaches reference the same training data when training the probability predictor. Moreover, input-based methods have the advantage of directly learning the counterfactuals (CFs) for each input, offering greater flexibility in representation at the cost of computational time. In comparing model-based methods, it's evident that VAEs struggle with considering differences in inputs or CFs. Therefore, it's not straightforward to declare one as superior to the other. In our paper, we focused on demonstrating that model-based methods, despite their lower computational time, can achieve performance on par with or better than input-based methods.

---

### Official Review · Reviewer_TXsm · 2023-10-29

**Soundness:** 2 fair
**Presentation:** 2 fair
**Contribution:** 2 fair
**Rating:** 3
**Confidence:** 4

**Summary:**

The is paper is concerned with extending the applicability of counterfactual explanation to tabular data, as well as increasing the computational efficiency of producing such explanations.  The authors approach is to propose a normalizing flow along with an ordinal variable encoding to account for cost of perturbations.

**Strengths:**

- The authors experiments help to identify why standard VAEs fail to produce a good amount of variation in their generated CEs.

**Weaknesses:**

- Experimentally, the authors omit a comparison to CeFLow of Duong et al 2023, which is the most natural comparator to their method (being itself a CF model for tabular data based on normalizing flows).  It's notable in its absence, how come?
- The authors explanation of how diversity is to be generated in CEs is too brief.  Section 3.2 very briefly states that the temperature parameter $t$ is used to add noise to the origins of the samples of $\mathbf{z}_{test}$.  But does this offer *more* diversity of valid samples than a method like DiCE, which intentionally penalizes the covariance between sampled points?  It's unsatisfying.
- The first paragraph in section 3 discusses some notation for CE generation, but states that:
> The aim of CE is to generate perturbed inputs $\mathbf{x}_{cf}$ such that $f(\mathbf{x}_{cf})>f(\mathbf{x})$ for the observed input x.

There are two mistakes here.  First is that  $>$ should be $!=$.  Second, it is far from settled what the benefits of CE are for, and is largely dependent on the person using the tool.  Users will take different insights from CEs than people building models, for example.  My reading of the paper of the paper suggests the authors are intending FastDCFlow to be a user-centric tool, so they should inform their perspective and their experiments accordingly.  Could they do a simulated user study, showing that FastDCFlow helps users make better decisions?  Or could they take an organizational risk perspective, showing that FastDCFlow helps an organization providing a model-based service make better (e.g fairer) decisions?

**Questions:**

- The second paragraph of the conclusion starts out with
> In subsequent phases of this study, we are currently integrating TE with a transformation technique that respects the order of categorical variables.

but section 6.1 suggests that TE was integrated for the experiment in this model, so I'm confused.  Is this just an error in tense?  Or does the present implementation of TE not respect order?  Or something else?
- The same paragraph of the conclusion continues
> Although predicted target values between the ML model evaluations using TE and OHE showed no marked differences, effectively adapting conversion methods for compact datasets remain a challenge.

I believe this statement needs way more unpacking.  If there were “no marked differences” between TE and OHE, I see two important questions that should be asked here:
1. Why is this so?  Is this a consequence of the evaluation criteria for CEs?
2. If OHE and TE don’t show a marked difference, what’s the justification for pursuing the integration of TE?

---

> ### Author Response · Authors · 2023-11-23
> **Thank you for your valuable review**
>
> Dear Reviewer TXsm,
>
> Thank you for your valuable feedback and suggestions on improving our paper. Here we address responses to your questions.
>
> At the beginning, we summarize significant revisions to this paper.
>
> **1) Modification in Implementation**
>
> Upon reviewing our source code, we discovered that for all model-based methods (CF_VAE, CF_CVAE, CF_DUVAE, FastDCFlow), the training data used was limited to samples where the ML prediction was < 0.5. However, the correct approach is to utilize the entire training dataset regardless of the predicted values. We have corrected this in our source code, retrained these models, and updated the results in our paper accordingly.
>
> **2) Revision of Experimental Goals**
>
> We have revised the manuscript to reflect our aim of developing a model that performs comparably to Input-based methods, with reduced computational time. Our objective is not to develop a model that excels in every evaluation metric but to create a model that is faster than Input-based methods while still maintaining balanced high performance across all metrics. While Input-based methods have the advantage of learning from each input and hence higher representational power, they are also time-consuming. We have incorporated the results of our revised experiments and discussed our findings in relation to the paper's objectives.
>
> **Q1.** , the authors omit a comparison to CeFLow of Duong et al 2023, which is the most natural comparator to their method (being itself a CF model for tabular data based on normalizing flows). It's notable in its absence, how come?
>
> **A1.** CeFlow proposes transformations by LabelEncoding and variational dequation [1] for categorical variables, and we determined that a fair comparison could not be made.
>
> **Q2.** The authors explanation of how diversity is to be generated in CEs is too brief. Section 3.2 very briefly states that the temperature parameter is used to add noise to the origins of the samples. But does this offer more diversity of valid samples than a method like DiCE, which intentionally penalizes the covariance between sampled points?
>
> **A2.** In our paper, we demonstrate experimentally that diversity can be enhanced in a simple manner without the need for explicit loss addition, as done in DiCE []. To further substantiate this result, we directly applied the diversity term used in DiCE to the CFs generated by FastDCFlow and tracked its progression with respect to temperature, as shown in Figure 4 of our paper. The results confirm that as the temperature increases, the diversity term of DiCE also improves. This indicates that our approach to diversity, though simpler, is effective and comparable to methods like DiCE that intentionally penalize the covariance between sampled points.
>
> **Q3.** First is that > should be !.
>
> **A3.** Thank you for your observation. Indeed, in the context of CEs, it is not always mandatory to satisfy constraints, hence we have revised the notation to reflect that it 'should' be so, rather than 'must'. This change more accurately represents the nature of constraints in counterfactual explanations.

---

> > ### Author Response · Authors · 2023-11-23
> > **Thank you for your valuable review**
> >
> > **Q4.** Could they do a simulated user study, showing that FastDCFlow helps users make better decisions? Or could they take an organizational risk perspective, showing that FastDCFlow helps an organization providing a model-based service make better (e.g fairer) decisions?
> >
> > **A4.** Thank you for suggesting additional methods to evaluate our approach. However, due to time constraints, it is challenging to obtain user feedback for this paper, so we focused on evaluating general performance metrics. Additionally, assessing how well FastDCFlow considers data fairness is not within the scope of our current proposal. This aspect has been added to the limitations section in Chapter 7, Conclusion, to acknowledge this and suggest potential areas for future research.
> >
> > On the other hand, it is possible to incorporate domain knowledge into FastDCFlow. For instance, we can set features that are not perturbable in reality (such as gender or race) or features that require a monotonic increase (like age), and impose constraints on their perturbation. Additional experiments regarding domain constraints have been conducted and are compiled in Appendix B. This demonstrates FastDCFlow's capability to consider domain-specific knowledge, further enhancing its applicability and effectiveness.
> >
> > **Q5.** section 6.1 suggests that TE was integrated for the experiment in this model, so I'm confused. Is this just an error in tense? Or does the present implementation of TE not respect order? Or something else?
> >
> > **A5.** My apologies for the confusion; it appears to be an error in the description. TE has already been implemented in our current model. We will ensure that the manuscript is revised to accurately reflect this.
> >
> > **Q6**. I believe this statement needs way more unpacking. If there were “no marked differences” between TE and OHE, I see two important questions that should be asked here: 1) Why is this so? Is this a consequence of the evaluation criteria for CEs?
> >
> > **A6.** The actual observation here is that there were no statistically significant differences in the predictions made by the probability predictors (ML models f) trained with TE and those trained with OHE. This observation is entirely unrelated to CEs. To prevent any misunderstanding, we have amended the conclusion section to clarify this point.
> >
> > **Q7.** If OHE and TE don’t show a marked difference, what’s the justification for pursuing the integration of TE?
> >
> > **A7.** The primary advantage of incorporating TE, beyond its ability to appropriately consider perturbations in categories, lies in its capacity to maintain dimensionality. This contributes to improved learning efficiency, making TE a valuable component in our model's architecture, despite the observed lack of significant difference from OHE in certain aspects. This dimensional consistency offered by TE is particularly beneficial in complex model architectures.
> >
> > **References:**
> >
> > 1. Emiel Hoogeboom, Taco Cohen, and Jakub Mikolaj Tomczak. Learning discrete distributions by
> > dequantization. In Third Symposium on Advances in Approximate Bayesian Inference, 2021.

---

### Official Review · Reviewer_KKcY · 2023-10-29

**Soundness:** 2 fair
**Presentation:** 3 good
**Contribution:** 1 poor
**Rating:** 3
**Confidence:** 4

**Summary:**

This paper lays out a method to obtain counterfactual explanations (CEs) of machine learning (ML) models, employing normalizing flows to generate candidate CEs. Target Encoding (TE) is utilized to maintain some level of ordinality amongst categorical features.

**Strengths:**

- Results are mostly on par with current state of the art, dependent on the criteria end users are seeking
- CF parameters are analyzed in a useful way
- Besides some typos, the paper is well written, with clear formatting and a logical structure

**Weaknesses:**

- The main problem in the paper is its lack of novelty. The two main contributions involve a) using a latent space model and b) converting categorical features to continuous mappings. The first has been proposed many times in the literature, mostly using VAEs. Normalizing flows have also been used before in this context, as the paper references. The transformation of categorical features to continuous features is not new either, and as such, I find the paper's novelty somewhat lacking.
- When considering the usual metrics proximity (P), validity (V) and run time (RT), the proposed method does not perform best across any one metric. The diversity metrics are questionable since they do not appear normalized, thus worse proximity is likely to promote better diversity (see questions).
- The results therefore rely heavily on the proposed metrics, CV and CS, which themselves are left highly unjustified in the text. I do not find these functions particularly compelling, since CS relies on CV, which itself relies on the diversity metrics proposed.

**Questions:**

1. Based on Table 1, Inner Diversity (ID) and Outer Diversity (OD) compute the average $\ell_2$ differences between CEs for one test point and between CEs across multiple test points. Why was diversity not considered via the angle between CE perturbations rather than the $\ell_2$ norm between raw CEs? Using the $\ell_2$ norm alone means larger diversity can be achieved through CEs with very bad proximity.
2. For the Bank dataset, FastDCFlow achieves proximity an order of magnitude worse than other methods (this is serious), and also achieves worse validity. Yet, FastDCFlow achieves an order of magnitude higher performance on the CS metric which is used for the final assessment of the methods. Proximity and validity are the two fundamental goals of counterfactual explanations, and in this case FastDCFlow fails on both fronts while being pushed as the method with the best overall score. Can the authors please provide further justification of the CS metric (mostly the diversity evaluation in the CV metric) in the context of the above example?

---

> ### Author Response · Authors · 2023-11-23
> **Thank you for your valuable review**
>
> Dear Reviewer KKcY,
>
> Thank you for your valuable feedback and suggestions on improving our paper. Here we address responses to your questions.
>
> **Q1.** The main problem in the paper is its lack of novelty. The two main contributions involve a) using a latent space model and b) converting categorical features to continuous mappings.
>
> **A1.** Our aim is not to propose novelty in the model or transformation method per se, but rather, we find novelty in the application of model-based counterfactual explanations.
>
> Regarding point a), previous model-based methods, as indicated by [1], have primarily been based on VAEs, which tend to generate identical CFs for different inputs, failing to capture diversity. To our knowledge, there are no precedents for applying Normalizing flows in model-based methods, and it is in this aspect that our work introduces an innovative application in the context of counterfactual explanations (CEs). While we utilize the RealNVP [2] structure, our contribution lies in the novel approach of introducing perturbations in the latent space through the reparametrization trick and modifications in the loss function.
>
> As for point b), many existing methods, as referenced in [3, 4], primarily apply OneHotEncoding or LabelEncoding for the transformation of categorical variables. We highlight in our paper that perturbing these encodings can compromise the diversity of predicted values. The idea of introducing TargetEncoding for the perturbation of categorical variables has not been discussed in the cited literature. Our work delves into the effectiveness of TE in the context of CEs, thereby contributing a novel perspective to this area.
>
> **Q2.** When considering the usual metrics proximity (P), validity (V) and run time (RT), the proposed method does not perform best across any one metric.
>
> **A2.** To clarify our objectives, we have made revisions to both the implementation and the goals of our study.
>
> **1) Modification in Implementation**
>
> Upon reviewing our source code, we discovered that for all model-based methods (CF_VAE, CF_CVAE, CF_DUVAE, FastDCFlow), the training data used was limited to samples where the ML prediction was < 0.5. However, the correct approach is to utilize the entire training dataset regardless of the predicted values. We have corrected this in our source code, retrained these models, and updated the results in our paper accordingly.
>
> **2) Revision of Experimental Goals**
>
> We have revised the manuscript to reflect our aim of developing a model that performs comparably to Input-based methods, with reduced computational time. Our objective is not to develop a model that excels in every evaluation metric but to create a model that is faster than Input-based methods while still maintaining balanced high performance across all metrics. While Input-based methods have the advantage of learning from each input and hence higher representational power, they are also time-consuming. We have incorporated the results of our revised experiments and discussed our findings in relation to the paper's objectives.
>
> **Q3.** Why was diversity not considered via the angle between CE perturbations rather than the ℓ2 norm between raw CEs? Using the ℓ2 norm alone means larger diversity can be achieved through CEs with very bad proximity.
>
> **A3.** We believe that the use of the ℓ2 norm is appropriate for our analysis. Firstly, it is important to recognize that proximity and diversity are not necessarily in a trade-off relationship. For instance, if the centroid of a CF set is distant from the test input, the performance in terms of proximity will deteriorate regardless of the diversity within the CF set. Therefore, situations where both proximity and diversity performances are poor can occur. Furthermore, the comparison of distances between raw CEs and the ℓ2 norm of perturbation vectors (the deviation from the original test input) is essentially equivalent. We have made modifications to the manuscript to clarify this fact more effectively.
>
> **Q4.** For the Bank dataset, FastDCFlow achieves proximity an order of magnitude worse than other methods (this is serious), and also achieves worse validity.

---

> > ### Author Response · Authors · 2023-11-23
> > **Thank you for your valuable review**
> >
> > **Q4.** For the Bank dataset, FastDCFlow achieves proximity an order of magnitude worse than other methods (this is serious), and also achieves worse validity.
> >
> > **A4.** Similar to A2, we have conducted a re-experimentation and updated the results accordingly. Regarding validity, although the relative performance of FastDCFlow remains lower compared to other models, the difference is not substantial, and we consider it not to be critical. We believe this demonstrates the robustness of FastDCFlow in maintaining competitive performance, despite slight variations in specific metrics.
> >
> > **Q5.** Proximity and validity are the two fundamental goals of counterfactual explanations.
> >
> > **A5.** We acknowledge that proximity and validity are not the sole fundamental goals of counterfactual explanations, hence our consideration of diversity and runtime as well. We believe the reason for this query stems from potentially misleading expressions in the introduction of our paper. Consequently, we have carefully revisited and revised the introduction to clarify our comprehensive approach that includes, but is not limited to, proximity and validity.
> >
> > **O6.** Can the authors please provide further justification of the CS metric (mostly the diversity evaluation in the CV metric) in the context of the above example?
> >
> > **A6.** The overall performance evaluation presented a significant challenge for us. In our paper, we defined combined metrics, CV and CS, but we realized that their arbitrary configurability could be problematic. Acknowledging this, we have decided to discontinue the performance comparison based on CV and CS, and instead, we have revised our approach to focus solely on individual evaluation metrics. This modification ensures a more objective and clear assessment of the models' performance.
> >
> > **Reference:**
> >
> > 1. Divyat Mahajan, Chenhao Tan, and Amit Sharma. Preserving causal constraints in counterfactual
> > explanations for machine learning classifiers. arXiv preprint arXiv:1912.03277, 2019.
> > 2. Laurent Dinh, Jascha Sohl-Dickstein, and Samy Bengio. Density estimation using real nvp. arXiv
> > preprint arXiv:1605.08803, 2016.
> > 3. Ramaravind K Mothilal, Amit Sharma, and Chenhao Tan. Explaining machine learning classifiers
> > through diverse counterfactual explanations. In Proceedings of the 2020 Conference on Fairness, Accountability, and Transparency, pp. 607–617, 2020.
> > 4. Tri Dung Duong, Qian Li, and Guandong Xu. Ceflow: A robust and efficient counterfactual ex-
> > planation framework for tabular data using normalizing flows. In Pacific-Asia Conference on
> > Knowledge Discovery and Data Mining, pp. 133–144. Springer, 2023.

---

### Meta-Review · Area_Chair_jfxs · 2023-12-10

**Metareview:**

The paper trains a normalizing flow to generate counterfactual explanations by perturbing the latent space. The main strengths of the paper lie on the fact that i) (like other deep generative models) FastDCFlow allows sharing parameters across factual observations and thus efficiently generating counterfactual explanations at test time; ii) the results are on par with similar, e.g., VAE approaches. However, as pointed out by the reviewers,  the main weakness of the paper is its novelty and contributions. I believe that the idea of using normalizing flows to generate counterfactual explanations is a good idea.  However, based on the empirical results, it is unclear what is really the added value, and thus the practical impact, of this paper compared to previous approaches. The empirical results, nor the explanation on the treatment of categorical explanations seem to back-up  a strong contribution to be at the standard of ICLR. I thus encourage the authors to revisit their paper to further strengthen the unique aspects of their approach, which I believe has the potential to go beyond the current "an application of normalizing flows to efficiently provide counterfactual explanations".

**Justification For Why Not Higher Score:**

N/A

**Justification For Why Not Lower Score:**

N/A

---

### Decision · Program_Chairs · 2024-01-16

Reject